# Molecular Mechanisms of the Interactions of *N*-(2-Hydroxypropyl)methacrylamide Copolymers Designed for Cancer Therapy with Blood Plasma Proteins

**DOI:** 10.3390/pharmaceutics12020106

**Published:** 2020-01-28

**Authors:** Larisa Janisova, Andrey Gruzinov, Olga V. Zaborova, Nadiia Velychkivska, Ondřej Vaněk, Petr Chytil, Tomáš Etrych, Olga Janoušková, Xiaohan Zhang, Clement Blanchet, Christine M. Papadakis, Dmitri I. Svergun, Sergey K. Filippov

**Affiliations:** 1Institute of Macromolecular Chemistry, Czech Academy of Sciences, Heyrovského sq. 2, 16206 Prague, Czech Republic; 2European Molecular Biology Laboratory, Deutsches Elektronen-Synchrotron, Notkestr. 85, 22607 Hamburg, Germany; 3Department of Chemistry, Lomonosov Moscow State University, Leninskie Gory 1-3, 119991 Moscow, Russia; 4Department of Biochemistry, Faculty of Science, Charles University, Hlavova 2030/8, 12840 Prague, Czech Republic; 5Fachgebiet Physik weicher Materie, Physik-Department, Technische Universität München, James-Franck-Str. 1, 85748 Garching, Germany; 6Harvard John A. Paulson School of Engineering and Applied Sciences, Harvard University, Cambridge, MA 02138, USA

**Keywords:** drug delivery, pHPMA, plasma proteins, polymeric nanoparticles, stealth effect

## Abstract

The binding of plasma proteins to a drug carrier alters the circulation of nanoparticles (NPs) in the bloodstream, and, as a consequence, the anticancer efficiency of the entire nanoparticle drug delivery system. We investigate the possible interaction and the interaction mechanism of a polymeric drug delivery system based on *N*-(2-hydroxypropyl)methacrylamide (HPMA) copolymers (pHPMA) with the most abundant proteins in human blood plasma—namely, human serum albumin (HSA), immunoglobulin G (IgG), fibrinogen (Fbg), and apolipoprotein (Apo) E4 and A1—using a combination of small-angle X-ray scattering (SAXS), analytical ultracentrifugation (AUC), and nuclear magnetic resonance (NMR). Through rigorous investigation, we present evidence of weak interactions between proteins and polymeric nanomedicine. Such interactions do not result in the formation of the protein corona and do not affect the efficiency of the drug delivery.

## 1. Introduction

Currently, the design of anticancer therapeutics is focused on creating new systems with low toxicity, e.g., those based on nanocarriers, enabling targeted delivery and high efficiency of the treatment [1]. Usually, it is assumed that immediately after being introduced into the blood, nanoparticles (NPs) are surrounded by plasma proteins that form a biological coating around the individual NPs, known as the protein corona [1,2,3,4,5,6,7,8,9,10]. Various characteristics of NPs (their size and shape, surface properties, etc.) have an influence on a formation mechanism of the protein corona [5,7]. In this respect, two categories of proteins can be distinguished: opsonins, which can increase the recognition of the reticuloendothelial system and, as a result, lead to a rapid elimination from the organism, and dysopsonins, which can improve the blood circulation time of NPs [9,10,11]. Therefore, when creating targeted drug delivery systems (DDS), knowledge about the protein–NP interaction mechanism can be crucial.

Up until now, the formation of a protein corona was described for NPs based on polystyrene, gold, silica, and poly(ethylene glycol) (PEG), and was frequently used for masking other NPs [3,4,6,12]. Another way to increase the lifetime of NPs in the bloodstream is to create a particle with “stealth” properties. The “stealth” effect can be described as a phenomenon of preventing the adsorption of proteins onto NPs. It allows for prolonging blood clearance and reduces non-specific cellular uptake [2,3,4].

The presence of strong or weak interactions between nanoparticles and proteins determines the possibility of the occurrence and the mechanism of the corona formation. The presence of a corona is an important factor in determining the effectiveness of a nanoparticle as a drug carrier [13]. The *N*-(2-hydroxypropyl)methacrylamide (pHPMA)-based copolymers are widely used for various purposes in biomedical applications, mostly as drug carriers [14,15,16,17,18]. They are water-soluble, non-toxic, non-immunogenic, and biocompatible. In aqueous solution, pHPMA-based copolymers bearing cholesterol (pHPMA-Chol) and the antitumor drug doxorubicin (Dox) form self-assembled NPs that have a high anticancer efficiency [15,16,17,18]. Due to the presence of a cholesterol fragment in the pHPMA structure, it induces supramolecular nanoparticle formation, and such DDS demonstrate slower blood clearance, more effective tumor accumulation, and enhanced antitumor activity of the polymeric drug. The main advantage of the above system is its simple preparation, easy self-organization, and an improved antitumor efficacy due to the enhanced permeability and retention (EPR) effect [15,16,17,18,19,20,21,22,23].

In previous work, we found that no protein corona was formed by human serum albumin (HSA) around the pHPMA-based NPs, and only a weak interaction between the HSA and NPs was detected using small-angle X-ray scattering (SAXS). These interactions are strongly hindered by the presence of Dox, which is distributed in the HPMA shell. Two possible mechanisms of the interaction were proposed: (i) HSA interacts with the NPs’ cholesterol groups, or (ii) HSA is restricted in the meshes formed by pHPMA chains [21,22].

In this work, we provide not only a thorough analysis of the mechanism of the interaction with HSA, but also investigate possible interactions of the pHPMA NPs with the blood proteins immunoglobulin G (IgG), fibrinogen (Fbg), and apolipoprotein (Apo) E4 and A1. Traditionally, experimental methods such as Isothermal Titration Calorimetry, Fluorescence Spectroscopy, UV-VIS Spectroscopy, Dynamic Light Scattering Spectroscopy, Circular Dichroism Spectroscopy, or Raman Spectroscopy have been used to study nanoparticle–protein interactions [2,3,12,23]. To study the nanoparticle–protein interactions in detail, we utilize a combination of nuclear magnetic resonance (NMR), small-angle X-ray scattering (SAXS), and analytical ultracentrifugation (AUC) [24,25,26,27,28,29,30]. These methods have different sensitivity and provide complementary information about the protein binding to a nanoparticle. Thus, high-resolution NMR spectroscopy allows one to detect the protein binding to a specific part of the nanoparticle, as demonstrated recently for the complexes of ubiquitin with Au NPs, where NMR described the binding of gold NPs to a specific domain of ubiquitin [24,25]. In contrast, SAXS can easily distinguish between the separate components and their complexes in solution due to additional cross-term contributions appearing in the scattering intensity. Moreover, low-resolution shapes of the solutes can be restored ab initio, and the scattering from available atomic models can be readily compared with the experimental SAXS data [26,27]. SAXS and NMR are highly complementary methods for studying protein solutions. In particular, NMR structures of the subunits can be used in the SAXS analysis of complexes using rigid body refinement [28]. AUC provides highly complementary information about the distribution of sedimentation coefficients of individual components in mixtures, which yields further valuable information in the analysis of protein corona formation [29,30].

## 2. Materials and Methods

### 2.1. Materials, Synthesis, and Characterization of the Copolymer

Synthesis and methods of characterization of the pHPMA-cholesterol NPs (Figure 1) are described in the Appendix A. The final copolymer had the following characteristics: *M*_w_ = 21,100 g/mol, *M*_n_ = 18,600 g/mol, *D* = 1.14, and content of cholesterol = 2.2 mol %. After the dissolution in the aqueous buffer, the hydrodynamic radius was 14 nm [15,16].

### 2.2. SAXS Measurements

Synchrotron SAXS experiments were performed at the EMBL beamline P12 (DESY, Hamburg, Germany) using a pixel detector (PILATUS 2M). The X-ray scattering images were recorded for a sample–detector distance of 4.1 m, using a monochromatic incident X-ray beam with two different wavelengths (λ = 1.24 Å and λ = 0.92 Å) covering the range of momentum transfers 0.03 nm^−1^ < q < 5.1 nm^−1^ (q = 4π sin θ/λ, where 2θ is the scattering angle). A full description of the experimental details is available in the Appendix A.

### 2.3. NMR Measurements

High-resolution ^1^H NMR spectra were recorded with a Bruker Avance III 600 spectrometer operating at 600.2 MHz (Bruker BioSpin, Rheinstetten, Germany). The ^1^H spin–spin relaxation times T_2_ of HDO were measured at 600.2 MHz using the CPMG18 pulse sequence with t_d_ = 5 ms. Every experiment was made with 16 scans, and the relaxation delay between scans was 100 s. The obtained T_2_ relaxation curves were monoexponential and the fitting process always made it possible to determine a single value of the relaxation time. The standard Bruker STD NMR pulse sequence STDDIFFESGP.3 with water suppression was used. An off-resonance at 20 ppm was used, and selective protein saturation was achieved by irradiating protein signals for 2 s with a spin-lock filter of 30ms.

### 2.4. Analytical Ultracentrifugation

The sedimentation analysis was performed using a ProteomeLab XL-I analytical ultracentrifuge equipped with an An50Ti rotor (Beckman Coulter Life Sciences, Indianapolis, IN, USA) at a 0.5 or 40 mg mL^−1^ HSA and 1 or 18 mg mL^−1^ pHPMA-Chol NPs total loading concentration in 0.05 M sodium phosphate and 0.15 M NaCl buffer pH 7.4 (PBS), which was also used as a reference. A full description of the experimental details for analytical ultracentrifugation measurements is available in the Appendix A.

## 3. Results

Three techniques were utilized to study pHPMA-Chol copolymer NPs, the plasma proteins (HSA, IgG, Fbg, Apolipoprotein E4, and A1), the blood plasma itself, and the polymer/protein mixtures. Firstly, the individual components, i.e., NPs and protein solutions, were separately analyzed by SAXS, and the examples of the scattering curves from different protein samples are shown in Figure 2 (see details in the Appendix A).

The SAXS data were used for ab initio shape reconstruction of the free polymer NPs and for comparison with computed scattering from the available high-resolution crystal structures of proteins, using the programs DAMMIN and CRYSOL, respectively [26]. For the proteins displayed in Figure 2, the results confirmed the monomeric state in solution. Next, SAXS experiments were performed on mixtures of proteins and polymers to check for possible interactions. In the absence of interactions between the polymers and proteins, the scattering from their mixture can be represented as a linear combination of the scattering curves from the two components with appropriate volume fractions; if complexes are present, such a representation would not fit the experimental data. For all analyzed samples, the scattering patterns were computed from the best-fitting mixtures using the program OLIGOMER, which yielded strong agreement with the experimental data (Figure 3) from the mixtures of individual proteins and free NPs (see details in the Appendix A) [31,32]. This finding clearly pointed to the absence of significant interactions between the investigated proteins and pHPMA-Chol NPs. A similar result was also obtained for the native blood plasma and pHPMA-Chol NPs, indicating that other proteins present in the plasma do not interact with the NPs either.

Absorbance and interference optical detection systems were used for AUC measurements, allowing for accurate monitoring of the sedimentation in real time and for the determination of the possible interactions between pHPMA-Chol NPs and human blood plasma proteins (Figure 4) [33]. Analyses of the mixed solutions of proteins and NPs did not show additional peaks in the case of HSA and IgG, even at a HSA concentration as high as 40 mg·mL^–1^, which represents its physiological level in human blood.

Conformational changes and internal motions are just as important for the function of biomolecules as their chemical structures. NMR is an experimental technique that provides a clear insight into the behavior of these systems on an atomic level.

Saturation Transfer Difference (STD NMR) is one of the handiest NMR methods for the detection of temporal ligand–protein interactions in solution through monitoring the signals of a ligand (with spectroscopic properties suitable for high-resolution studies) regardless of the protein size and structure [34,35]. In our case, a part of the copolymer (located on the surface of the NPs) and cholesterol (located not deeply in the structure of NPs) possesses the necessary spectroscopic properties and allows us to explore the nature of the interaction of NPs with the protein. The presence of both a polymer and cholesterol signal on the STD spectrum (red spectrum, Figure 5) confirms the presence of a notable interaction between the HSA and the NP. Comparison of samples with different ratios of protein and NP concentrations suggests that enhancement of STD can only be explained by a saturation transfer between the protein and the NP. The STD enchantment for the NP in the presence of other proteins (Fbg, Apo A) can be neglected due to a weak signal comparable to the signal-to-noise ratio.

## 4. Discussion

Two possible mechanisms of the interaction were proposed: (i) HSA binds to the NPs’ cholesterol groups, or (ii) HSA is restricted in the meshes formed by pHPMA chains [21]. The finding from SAXS measurements and fitting clearly pointed to the absence of significant interactions between the investigated proteins and NPs. A similar result was also obtained for the native blood plasma and NPs, indicating that other proteins present in the plasma do not have a strong interaction with the NPs either. Some weak interactions between NPs with Fbg and Apo E4 were represented by a slight shift of the sedimentation coefficient distribution and the presence of an additional peak, respectively (Figure 4). However, for Apo A1, the whole distribution changes upon mixing it with NPs and resembles the shape of the distribution of free NPs, confirming the presence of weak cholesterol–protein interactions. Spin-spin relaxation times (T_2_) measured by NMR provide the dynamic picture of the segment’s movement on a picoseconds–nanoseconds time scale. This allows for characterization of the motions of flexible polymer segments and solvent separately [36,37]. The spin relaxation time T_2_ is sensitive to both the local molecular dynamics and the local density of protons in the medium [37]. The effect of the component in the mixture can be evaluated using the T_2_ ratio from neat NPs or protein in comparison to their mixture. The ratio, approximately equal to 1, characterizes the absence of changes in the behavior of the components in the mixture in comparison with the pure sample. Increasing the value of the ratio T_2_^HDO^(HSA)/T_2_^HDO^(HSA+NP) assumes the presence of some kind of interaction that affects the mobility of water in the solution. Moreover, we followed the changes in the pHPMA methyl group (which remains mobile on the surface of the nanoparticle), and no restriction in mobility due to the pHPMA and HSA interaction was observed (Table 1). This result suggests that the interaction of the NP with HSA does not go through the HPMA loop and can be an argument in favor of the second hypothesis that the interaction occurs between HSA and cholesterol.

The measurement of relaxation times does not confirm that HSA is restricted in meshes formed by pHPMA chains, but it also does not provide information about the interaction of protein with NPs via cholesterol. However, by changing the saturation time in STD NMR experiments, we can evaluate the influence of the polymer and cholesterol signals on the STD amplitude separately.

STD NMR profiles (Figure 6) for pHPMA-Chol copolymer NPs in the presence of HSA demonstrate a more intense effect on the cholesterol in comparison with the STD enhancement on the pHPMA signals. This fact proves the mechanism of the interaction between protein and NPs through cholesterol.

## 5. Conclusions

The presence of strong or weak interactions between nanoparticles and proteins determines the possibility of occurrence and the mechanism of corona formation. The presence or absence of a corona is an important factor in determining the effectiveness of a nanoparticle as a drug carrier. The complementary methods SAXS and AUC witness that no thick, hard, or soft protein corona from HSA, IgG, Fbg, Apo E4, Apo A1, or plasma itself is formed around pHPMA-Chol copolymer NPs. This result proves the perfect “stealth” properties of pHPMA without any absorption of protein on the nanocarriers designed for the cancer therapy. Meanwhile, the results from a combination of SAXS, AUC, and NMR demonstrate the existence of weak interactions between proteins (HSA and Apo A1) and cholesterol groups of NPs. However, these interactions do not hamper the drug delivery potency of the studied pHPMA-Chol copolymer NPs.

## Figures and Tables

**Figure 1 pharmaceutics-12-00106-f001:**
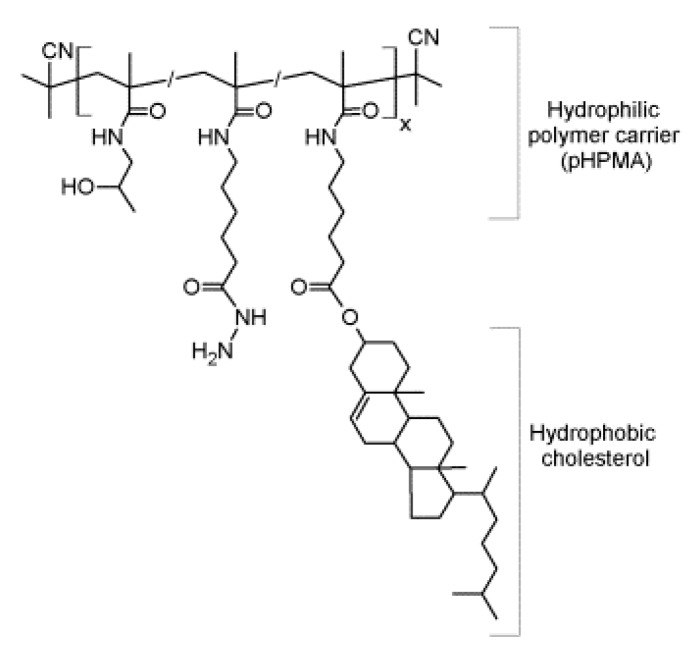
Chemical structure of the *N*-(2-hydroxypropyl)methacrylamide (HPMA)-based polymers bearing cholesterol (pHPMA-Chol) copolymer: *M*_w_ = 21,100 g/mol; content of cholesterol = 2.2 mol %; *D* = 1.14.

**Figure 2 pharmaceutics-12-00106-f002:**
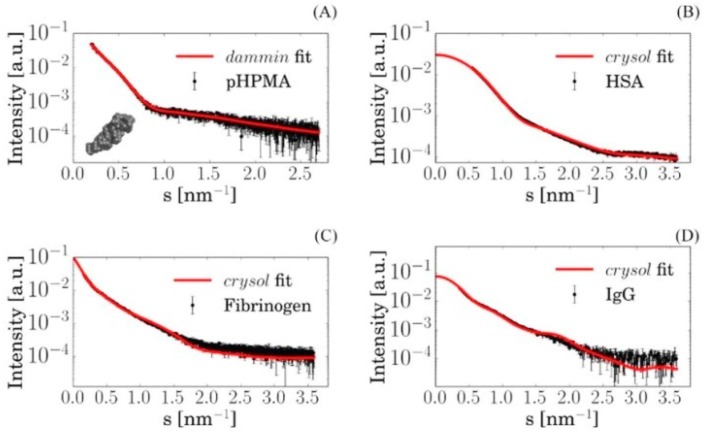
Small-angle X-ray scattering (SAXS) data from solutions of different proteins (dots) and corresponding fits (solid lines) of the high-resolution Protein Data Bank (PDB) model using CRYSOL and ab initio shape reconstruction using DAMMIN, respectively: (**A**) pHPMA nanoparticles (NPs) (inset shows the ab initio shape reconstruction model); (**B**) human serum albumin (HSA); (**C**) fibrinogen (Fbg); (**D**) immunoglobulin G (IgG).

**Figure 3 pharmaceutics-12-00106-f003:**
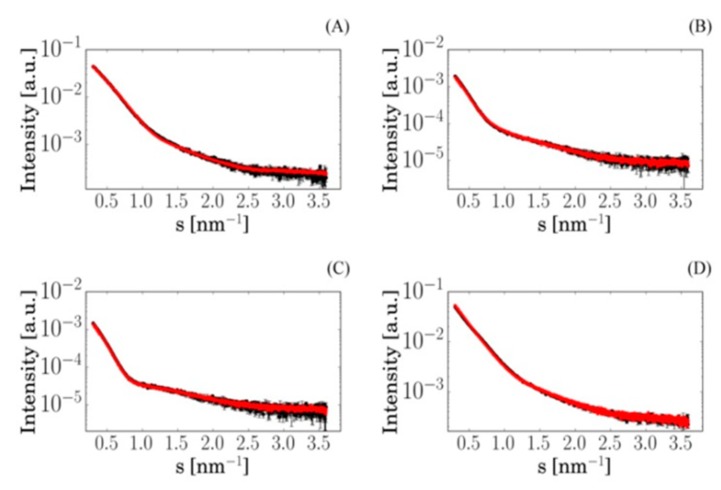
SAXS curves from mixed solutions of pHPMA-Chol copolymer NPs with proteins at different polymer/protein ratios (experimental curve: black). Each curve was modeled as a linear combination of the curves from the two pure components measured separately (calculated curve: red): (**A**) pHPMA-Chol/HSA = 18/20, (**B**) pHPMA-Chol/IgG = 18/5, (**C**) pHPMA-Chol/Fbg = 18/1, and (**D**) pHPMA-Chol/plasma = 18/20.

**Figure 4 pharmaceutics-12-00106-f004:**
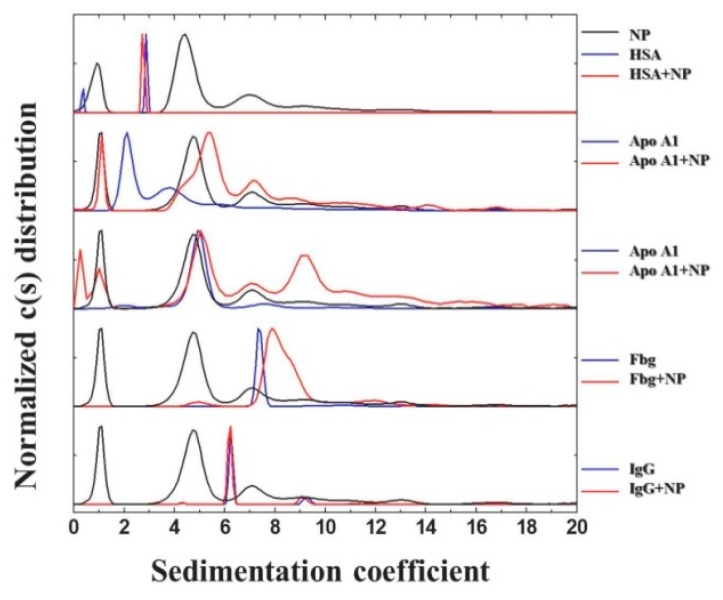
Analytical ultracentrifugation (AUC) analysis of protein binding to pHPMA-Chol NPs. Overlaid normalized sedimentation coefficient distributions of the sedimenting species are shown for individual proteins (blue lines), NPs (black line), and their mixed solutions (red lines).

**Figure 5 pharmaceutics-12-00106-f005:**
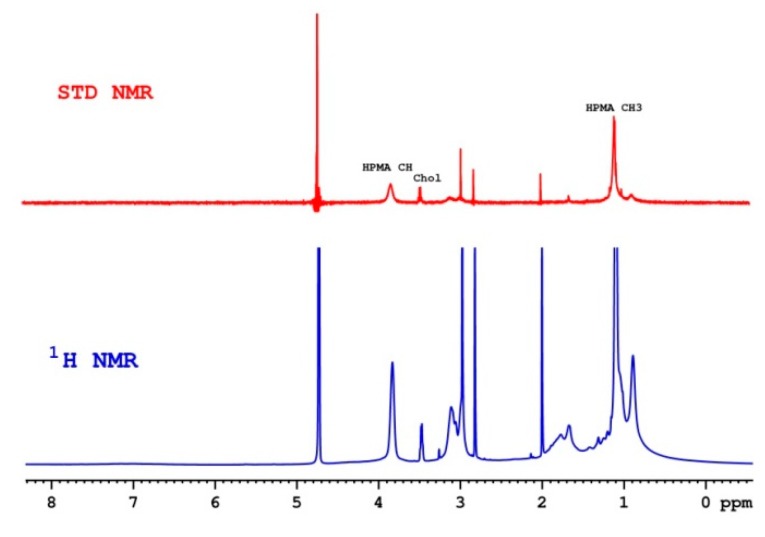
^1^H nuclear magnetic resonance (NMR) spectrum (blue) of pHPMA-Chol in the presence of HSA and the corresponding Saturation Transfer Difference (STD NMR) spectrum (red).

**Figure 6 pharmaceutics-12-00106-f006:**
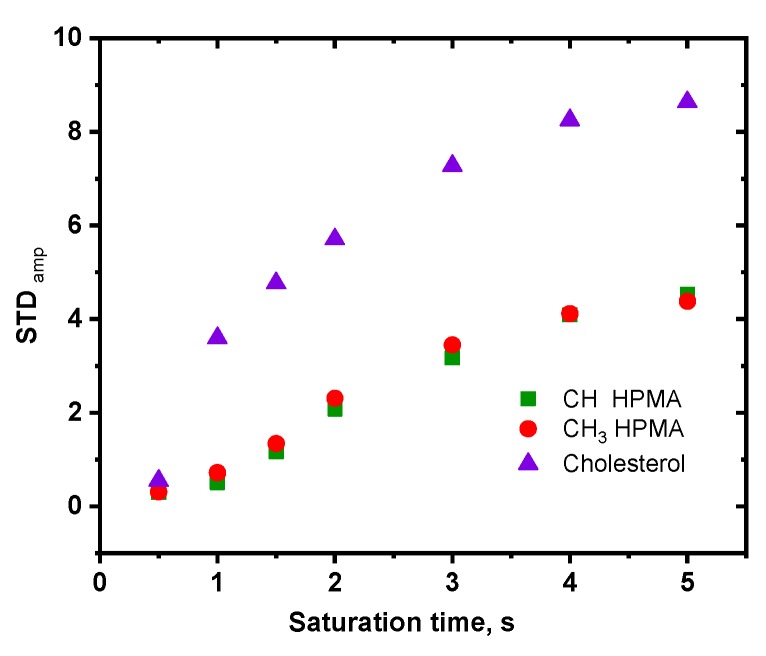
STD NMR profiles for pHPMA-Chol copolymer NPs in the presence of HSA for various protons of pHPMA-Chol copolymer.

**Table 1 pharmaceutics-12-00106-t001:** Ratios of proton relaxation time (T_2_) for HDO (solvent) molecules and pHPMA-Chol NPs.

c(HSA) mg·mL^−1^		T_2_^HDO^(HSA)/T_2_^HDO^(HSA+NP)		T_2_^HPMA^(NP)/T_2_^HPMA^(HSA+NP)	
		2.5	35/40	2.5	35/40
**c(pHPMA-Chol) mg/mL**	**4**	1.24	1.48/-	1.1	1/-
	**18**	3.3	-/4.1	1.3	-/1.1

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
