# Peer review of "Molecular Mechanisms of the Interactions of N-(2-Hydroxypropyl)methacrylamide Copolymers Designed for Cancer Therapy with Blood Plasma Proteins"

_pharmaceutics, 2020, doi:10.3390/pharmaceutics12020106_

Round 1
Reviewer 1 Report
In this manuscript, the authors investigated the interaction between pHPMA and several model blood plasma proteins by SAXS, NMR and AUC. The results showed that the pHPMA materials exhibited good antifouling properties and there were weak collisional interactions between proteins and the NPs formed by pHPMA copolymers. The results were beneficial for other researchers using pHPMA for their drug delivery work. There are some minor problems with this manuscript.
For the polymer/protein interaction, the authors don’t include PEG, zwitterionic materials, or some other fouling or antifouling materials as control for comparison. The authors are investigating the interactions between pHPMA polymers and plasma proteins (as claimed in title). Why do they use nanoparticles instead of bulk polymeric materials (hydrogels or membranes) with some more quantitative and accurate methods? The impact of this work is not very high with the obtained results.Author Response
We thank the reviewer for this valuable comment. The formation of protein corona for bulk PEG and PEGilated nanoparticles was already reported [5b, 10]. It was proven that PEG interacts with blood plasma proteins. Similarly, hydrogels or membranes made of polymers with hydrophobic and charged groups also interact strongly with blood plasma proteins. Nevertheless, polymer properties in bulk and solution could be different. There were numerous reports that, for example, the values of binding constants for surface immobilized polymer measured by ELISA are significantly different from the binding constants measured by isothermal titration calorimetry (ITC) for polymers in solution. All the polymer materials we investigated are tailored for the drug delivery purposes, thus we are working with nano-sized objects, which are suitable for this application. Neither hydrogels, nor membranes are not suitable for this application. The presence or absence of protein corona is an important factor determining the effectiveness of a nanoparticle as a drug carrier. Knowledge of the NP- individual protein interaction can be useful for enhancing drug delivery effectivity. From our point of view, we believe that the impact of this work is rather high due to biologic relevance of the systems under the study.
Reviewer 2 Report
This work conducted by Janisova et al. reports the molecular mechanisms interaction of N-(2-hydroxypropyl) methacrylamide copolymers with the blood plasma proteins. The authors have used a range of analytic tools, including SAXS, NMR and AUC, to characterize such interactions. Their results proved the “stealth” properties of pHPMA with little protein adsorption activity, and that the weak collisional interactions is through cholesterol. In my opinion, the work provides some new knowledge in the corona formation on nanoparticles, but the experimental condition is too simple compared to the real in vivo condition. Therefore, I would recommend the acceptance of this manuscript for publication after a major revision.
Dynamic light scattering analysis is recommended to characterize the hydrodynamic size and aggregation of nanoparticles after co-incubation with proteins. The work studied the mixture of NPs with proteins individually. However, in the in vivo condition, hundreds types of proteins are mixed together, providing a highly complex environment, and the interaction between the proteins and NPs can be altered compared to the simple condition investigated in the present work. Therefore, it would be more meaningful to study the protein/NPs interactions under a condition that contains a series of types of proteins rather than only one. There is no Figure 2A inset.
Reviewer 3 Report
In the manuscript entitled “Molecular mechanisms of the interactions of N-(2-hydroxypropyl) methacrylamide copolymers designed for cancer therapy with the blood plasma proteins” the authors investigate the interactions between N-(2-hydroxypropyl)methacrylamide (HPMA) polymeric nanoparticles with proteins in human blood plasma. The overall conclusion is that blood plasma proteins form no hard corona around HPMA NPs. It is the same conclusion that some of the authors of the present manuscript reach in a previous published paper (Doi: 10.1039/C7NR09355A, Nanoscale, 2018, 10, 6194). However, they come to the same conclusion by using different methods. It is important to indicate similar work, competing methods and alternatives to the present work, in order to justify the novelty of this manuscript. How broad are the conclusions presented in this manuscript in relation to the existing literature? How likely are other groups to adopt the methods or approaches introduced in this manuscript? The introduction section should discuss existing studies on the interaction of polymeric nanoparticles and plasma proteins, including the same article (Nanoscale, 2018, 10, 6194) previously published. There is also some references missing, such as this one, doi: 10.1038/s41467-017-00600-w; Nat Commun 8, 777 (2017), where the authors used a library of polymer nanoparticles to show how physicochemical characteristics influence blood circulation and early distribution.
The manuscript is clearly written, and the results are well presented. The results appear to be valid and the methodology is appropriate.
Author Response
We thank reviewer for the comments. We completely agree with the reviewer that the introduction section should contain the discussion of the existing studies on the interaction of polymeric nanoparticles and plasma proteins. For this purpose, we have expanded the review of recent studies on this topic, including the articles recommended by the reviewer. The effectiveness of drug carriers is determined by their interaction with blood proteins. In this paper, we study the effect of not only the characteristics of the nanoparticles as a whole we propose a method for studying the interaction of individual constituents of a nanoparticle with proteins. A detailed study of such interactions may allow tailoring the nanoparticles with specified parameters for the particular purposes. In previous works, we have found that the HSA-NPs interactions are strongly hindered by the presence of drug (doxorubicin), which is distributed in the HPMA shell [13c, 16b]. Investigation of the possible mechanism of NP-protein interaction is the aim of research of this article. Thus we believe that the investigation of this mechanism can be highly interesting even the polymer do not contain the drug.
Reviewer 4 Report
The manuscript entitled “Molecular mechanisms of the interactions of N-(2-hydroxypropyl) methacrylamide copolymers designed for cancer therapy with the blood plasma proteins” presents the fabrication of polymeric nanoparticles and their ability of stealth effect along with proposed mechanistic elucidations based on SAXS, STD NMR, and AUS data and other investigations. The article could be publishable after addressing the following comments.
Detailed introduction of the novelty and outline of the study should be given at the end of the introduction for the reader’s convenience.
I suggest the author move the details of supporting info to the main text.
Although the materials were characterized well, morphological attributes, as well as chemical functionalities of fabricated pHPMA-Chol copolymer NPs, should be presented.
The authors stated that “Such collisional interactions have not resulted in the formation of the protein corona and do not affect the efficiency of the drug delivery.” I would suggest demonstrating the effect of protein corona on drug delivery using a drug model.
Abbreviations should be defined when they appear for the first time in the abstract and Main text.
Minor Typo errors, in terms of punctuations and grammar related, were evident throughout the manuscript.
Figures are of inferior quality, better edit them appropriately for a better view of the reader. In addition, different Font styling in the same figure (for instance, Figure 4). Unify them all.
Author Response
We thank the reviewer for the valuable comments. The aim of the study was based on the proper analysis of protein corona forming around the NPs and NP-HSA interactions intended for the drug delivery purposes. Thus we have not described the attributes in more details. The characterization of polymer NPs is described in the chapter 2.1. and SI. With the aim to increase the readability we have added the references to the papers describing the morphological and chemical composition in more details. We have expanded the Introduction section by the review of recent studies on this topic and outline of the study, adjusted the use of abbreviation, recommended by the reviewer.Recently, we have investigated both NPs, Dox-free and Dox containing. We have found quite similar results. Moreover, we have found that the HSA-NPs interactions is less pronounced in the case of Dox containing polymer NPs. When Dox is absent, HSA could binds to the cholesterol groups which form the core of the NPs by diffusing through the loose HPMA shell, or gets caught in meshes formed by the HPMA chains. These interactions are strongly hindered by the presence of Dox, which is distributed in the HPMA shell, meaning that the delivery of Dox by the NPs in the human body is not affected by the presence of HSA [16b]. Study of the mechanism of this interaction is the aim of research of this article. We believe that the results of the paper on Dox-free NPs are highly interesting even the polymer do not contain the drug.
Round 2
Reviewer 2 Report
I'm satisfied with the changes made by authors.
Reviewer 3 Report
The authors have addressed all the comments and suggestions I made in the first review. The quality of the article has significantly improved, particularly the introduction now clearly shows what the authors had achieved previously and what they want to achieved now. I hope the same methodology will be applied to other studies in the future.